# Genome-Wide Identification and Transcriptome-Based Expression Profile of Cuticular Protein Genes in *Antheraea pernyi*

**DOI:** 10.3390/ijms24086991

**Published:** 2023-04-10

**Authors:** Xin Fu, Miaomiao Chen, Runxi Xia, Xinyu Li, Qun Li, Yuping Li, Huiying Cao, Yanqun Liu

**Affiliations:** 1College of Bioscience and Biotechnology, Shenyang Agricultural University, 120 Dongling Road, Shenyang 110866, China; 2Research Group of Silkworm Breeding, Sericultural Institute of Liaoning Province, Liaoning Academy of Agricultural Sciences, 108 Fengshan Road, Fengcheng 118100, China

**Keywords:** *Antheraea pernyi*, cuticular protein, epidermis, non-epidermis, expression profile, transcriptome

## Abstract

*Antheraea pernyi* is one of the most famous edible and silk-producing wild silkworms of Saturniidae. Structural cuticular proteins (CPs) are the primary component of insect cuticle. In this paper, the CPs in the genome of *A. pernyi* were identified and compared with those of the lepidopteran model species *Bombyx mori*, and expression patterns were analyzed based on the transcriptomic data from the larval epidermis/integument (epidermis in the following) and some non-epidermis tissues/organs of two silkworm species. A total of 217 CPs was identified in the *A. pernyi* genome, a comparable number to *B. mori* (236 CPs), with CPLCP and CPG families being the main contribution to the number difference between two silkworm species. We found more RR-2 genes expressed in the larval epidermis of fifth instar of *A. pernyi* than *B. mori*, but less RR-2 genes expressed in the prothoracic gland of *A. pernyi* than *B. mori*, which suggests that the hardness difference in the larval epidermis and prothoracic gland between the two species may be caused by the number of RR-2 genes expressed. We also revealed that, in *B. mori*, the number of CP genes expressed in the corpus allatum and prothoracic gland of fifth instar was higher than that in the larval epidermis. Our work provided an overall framework for functional research into the CP genes of Saturniidae.

## 1. Introduction

Structural cuticular proteins (CPs) are the primary component of insect cuticle [1,2], and can form the Bouligand model to stabilize the complex structure of the cuticle in maintaining the elasticity and other physical properties of insects [3]. Evidence from *Culex pipiens pallens* L. [4,5], *Aedes aegypti* L. [6] and *Bactrocera dorsalis* Hendel [7] have indicated that some CPs are involved in the response to pesticides and arid and extreme temperature conditions. However, CPs vary greatly in number, sequence and structure within and between species [7,8,9,10].

In insects, more than a dozen CP families have been identified according to their conserved motifs [11]. The largest CP family is CPR, which contains the chitin-binding domain called the Rebers and Riddiford motif (R&R) [12,13,14]. CPs in the CPR family are further assigned into three subfamilies, RR-1, RR-2 and RR-3 [15,16,17,18]. Most of the remaining CP families contain conserved motifs, including cuticular proteins analogous to peritrophins (CPAPs). CPAPs are also classified into the CPAP1 or CPAP3 subfamily according to the number of the ChtBD2 (chitin-binding) domain contained [19]. Other CP families containing conserved motifs include the CPF family, which has a 44 aa (amino acid) consensus [16,20]. The CPF-like (CPFL) family has a C-terminal region similar to CPF but lacking the 44 conserved aa residues [17]. The CPT has a Tweedle motif [21], the CPCFC has two or three repeats of the C-X_5_-C motif [11,22] and the 18 aa has an 18 amino acid motif [18,23]. The CPLCA and CPLCP are low-complexity families with their own distinct sequence features [24]. Some CPLC families were also identified without conserved motifs as being rich in glycine or as containing repeats of AAP (A/V), P (V/Y), GYGL or GLLG. The CPG family was first identified in the model insect *Bombyx mori* L. (Lepidoptera: Bombycidae) with multiple GGYGG or GGxGG repeats, and the CPH family comprised hypothetical CPs [10,11,25]. These findings facilitate the accurate identification and classification of candidate structural CPs in insects from genomic and transcriptomic data.

So far, knowledge on the CPs of wild silkworms is severely limited. Wild silkworms of the family Saturniidae contain ~3400 species [26], and some wild silkworms play an important role in international textile commerce and local edible insect resources [27]. *Antheraea pernyi* Guérin-Méneville (Lepidoptera: Saturniidae) is one of the most famous wild silkworms used for edible insect resources and wild silk production. The yield of cocoons (pupae) of *A. pernyi* is only second to *B. mori*. Unlike *B. mori*, whose larvae are reared indoors, the larvae of *A. pernyi* are still reared in the field. Morphologically, *A. pernyi* exhibits colorful and relatively harder larval epidermis/integument (epidermis in the following), while *B. mori* shows white, opaque and softer larval epidermis (Appendix A). For the larval internal prothoracic gland, an important non-epidermis tissue/organ, *B. mori*, has a relatively solid and intact structure that is difficult to be broken in dissection, while *A. pernyi* shows a soft and loose structure that is easy to be broken in dissection (Appendix A). Although the CP genes of *B. mori* have been well studied [12,24,25,28,29,30], the CP genes of *A. pernyi* have not been identified at the genome level. Recently, the chromosomal-level genome sequence of *A. pernyi* has been released [31], which lays a foundation for identifying the CP genes for this species.

In the present study, we used the recently published genome of *A. pernyi* to identify its CPs and we compared the CPs of *A. pernyi* with those of *B. mori*. The transcriptomic data were further used to compare the expression pattern of CP genes in the epidermis and some non-epidermis tissues/organs within and between species. The objectives of this study were as follows: (i) to identify and characterize the CPs of *A. pernyi*; (ii) to ascertain the mechanism underlying the hardness differences in the larval epidermis and prothoracic gland between two silkworm species by comparing the expression pattern of CP genes. The results will establish an overall framework of information for further functional research into the CP genes of the Saturniidae species and provide some insights into the CPs of insects.

## 2. Results

In the first step, the CP genes of *B. mori* were re-identified based on the latest released high-quality genome assembly (Genbank accession number: GCF_014903235.1) [32]. A total of 220 CP genes were identified in the first draft genome of *B. mori* [25]. Subsequently, 28 CP genes were identified, including 4 CPR genes, 15 CPAP1 genes and 9 CPAP3 genes [12,29]. These CP genes were defined into 12 families (CPR, CPT, CPAP1, CPAP3, CPF, CPFL, CPCFC, CPLCP, CPLCA, CPG, 18aa and CPH) [12,24,25,28,29,30]. Using BLASTp against the latest genome assembly [32], a total of 236 previously reported CP genes could be retrieved; the remaining 12 CP genes could not be found (Appendix A). Thus, these 236 CP genes in the latest genome assembly of *B. mori* were used in the following analysis. Cuticular protein locations on chromosomes are given in Appendix A.

In the chromosome-scale genome of *A. pernyi* [31], a total of 217 gene models belonging to nine CP groups (157 CPR, 18 CPAP, 1 CPF, 2 CPFL, 4 CPT, 3 CPCFC, 2 CPLCP, 8 CPG and 22 CPH) were identified (Table 1), accounting for 1% of the annotated genes in this species. The vast majority of CP genes were full-length. The CP genes of *A. pernyi* from the CPAP1, CPAP3, CPF, CPFL, Tweedle and CPCFC families were named according to the orthologous *B. mori*, while the other CP genes were named in the order in which they were annotated (Appendix A). Cuticular protein locations on chromosomes are given in Figure 1.

### 2.1. CPR Family

As the most abundant CP family, CPR has the largest number in most insects [12,14], and this case was also observed in *A. pernyi.* In total, 157 of the annotated genes containing an R&R consensus region were identified in the genome of *A. pernyi* and were assigned to the CPR family, comprising ~71% of the CP genes identified. Among them, 52 and 92 were further classified into the RR-1 and RR-2 subfamilies, respectively, based on the combination of the CuticleDB website and the phylogenetic results. We found that GWHPABGR004082 in *A. pernyi* shared 69% sequence identity with BmorCPG21 (Appendix A). Sequence alignment results showed that both of them had abundant glycine, but the result of CuticleDB classified GWHPABGR004082 into the RR-1 subfamily. In *B. mori*, BmorCPR41–46 genes are flanked by BmorCPG21 on one end and ORC3 and the T-related protein on the other end [28]. We then assigned GWHPABGR004082 as the CPR family. Based on sequence similarity to four known RR-3 proteins in *B. mori*, two RR-3 proteins were identified in *A. pernyi*. The remaining 11 sequences that were not assigned to any of the previous groups were classified as CPRNC (CPR not classified).

The analysis of genomic organization revealed that the vast majority of CPR genes of *A. pernyi* were located in large tandem arrays. Many genes appeared in the form of gene clusters: 39 of the RR-1 genes were found in tandem arrays of 5–21 genes on four chromosomes (chr18, chr26, chr33 and chr34), while 79 of the RR-2 genes were found in tandem arrays of 6–52 genes on four chromosomes (chr14, chr17, chr18 and chr46). This was similar to the CPR family in other lepidopteran species [7]. In particular, RR-1 and RR-2 genes were not found to co-exist within the same tandem array, similar to that previously reported in *Anopheles gambiae* Giles [8] and *Anopheles sinensis* Wiedemann [33].

According to the phylogenetic analysis, we were able to assign orthologs to 34 of 51 *B. mori* RR-1 genes. It was shown that the homology of the RR-1 family is clear (Figure 2A). WebLogo analysis [34] confirmed that RR-1 proteins in *A. pernyi* contain the conserved R&R Consensus region (GxFxYxxPDGxxxxVxYxADENGYQPxGAHLP) (Figure 2B). However, determining orthologs among RR-2 proteins was difficult, as only 31 of 84 *B. mori* RR-2 genes had clear 1:1 orthologs in *A. pernyi* (Figure 3A). The remaining RR-2 genes have no clear orthologous genes. Sequence logos generated from 176 RR-2 genes of *A. pernyi* and *B. mori* confirmed that RR-2 proteins in *A. pernyi* contain the conserved R&R Consensus region (EYDAxPxYxFxYxDxHTGDxKSQxExRDGDVVxGxYSLxExDGxxRTVxYTADxxNGFNAVVxxE) (Figure 3B). Some clades in RR-2 proteins showed multiple to multiple homologous relationships (co-orthologous groups) in our BLAST analysis (Appendix A). We further compared these co-orthogonal groups with the results in [25], and found that they corresponded to the gene clusters in *B. mori* and had specific locations on chromosomes (Figure 3A). For example, six genes (ApCPR113, 117, 118, 119, 120 and 121) clustered on the same chromosome chr18 were the most similar to BmCPR86–89. In Appendix A, the similar homologous groupings are shown.

### 2.2. CPAP Superfamily

A total of 12 CPAP1 genes and 6 CPAP3 genes were identified in *A. pernyi* by the sequences Cx_14–16_Cx_5_Cx_9–13_Cx_12_Cx_7–8_C and Cx_13–24_Cx_5_Cx_9–10_Cx_12–16_Cx_7–8_C, similar to the number found in *B. mori* (14 CPAP1 and 9 CPAP3 genes, respectively). Sequence alignment clearly showed the difference between CPAP1 and CPAP3 (Appendix A). We named *A. pernyi* CPAP genes based on orthologs to *B. mori* and homologous genes in other insects. Note that each CPAP3-A and CPAP3-D contained two paralogs (CPAP3-A1, A2 and CPAP3-D1, D2) in *B. mori* [28], while only ApCPAP3-A1, ApCPAP3-A2 and ApCPAP3-D2 were identified in *A. pernyi*. All six ApCPAP3 genes were located on chr4, and three of them (A1, A2 and B) were present in tandem arrays. CPAP1 genes did not exhibit the same clustering characteristic, and they were distributed on nine different chromosomes. The distribution pattern of the CPAP family genes in *A. pernyi* was the same as observed in *B. mori* [28].

The phylogenetic analysis showed that the ChtBD2 domains from CPAP1 and CPAP3 proteins formed two distinct branches (Figure 4A). One branch contained all ChtBD2 domain sequences belonging to the CPAP1 family, while the other branch contained three ChtBD2 domain sequences of the CPAP3 family. Their domains are shown in Figure 4B. In total, 10 CPAP1 genes and 6 CPAP3 genes in *A. pernyi* appeared to be 1-1 orthologs to *B. mori*, with amino acid sequence identities above 49%. ApCPAP3s shared high amino acid sequence identities (81–96%) with BmorCPAP3s (Appendix A). Three CPAP1 genes (CPAP1-G/H/N) and three CPAP3 genes (CPAP3-D1/E2/E4) were present in *B. mori* but absent in *A. pernyi*.

### 2.3. Other CP Families

In *A. pernyi*, only one gene (GWHPABGR002600) belonged to CPF. This case was the same as found in *B. mori* [25]. ApCPF and BmorCPF were 1-1 orthologous genes with a high sequence identity of 81% (Appendix A). Two proteins GWHPABGR002511 and GWHPABGR002513 were identified as CPFLs (CPF-like) in *A. pernyi*, and both of them were homologous to BmorCPFL4 (Appendix A).

Four CPT genes with the Tweedle motif were identified in *A. pernyi*, as found in other lepidopteran species [35]. The four ApCPTs were found to be homologous to BmorCPT1-4 (Figure 5A). Sequence logos generated from eight CPTs of *A. pernyi* and *B. mori* revealed four well-conserved blocks (Figure 5B). Block I contained a KX_2_Y/F motif (where X_2_ represents two non-conserved aas), II showed a KX_4-5_FIKAP sequence, III had a TX_2_YVL motif and IV consisted of the KPEVYXFV/IKY sequence.

The members of the CPCFC family are generally present in 1–2 copies per species in insects [22,30]. We identified three members of the CPCFC family in *A. pernyi*, which was different from only one CPCFC protein in *B. mori* (Appendix A). Weblogo analysis confirmed two repeats of C-X_5_-C in *A. pernyi* CPCFC proteins (Figure 6A). Three CPCFCs in *A. pernyi* shared more than 59% sequence identities with *B. mori* CPCFC (BmorCPH1) (Appendix A).

In *A. pernyi*, the CPLCA family was not found; however, two CPLCP genes were identified, which contained 29 occurrences of PV and 30 occurrences of PY. WebLogo analysis showed a high density of prolines with PV and PY sequences in these proteins (Figure 6B). Via sequence comparison, we reassigned six members (CPG7/8/11/12/13/24) of *B. mori* as the CPLCP family. ApCPLCP1 shared 80% amino acid sequence identity with BmorCPG24, and ApCPLCP2 shared 58% amino acid sequence identity with BmorCPG8.

So far, the CPG family proteins have only been found in the lepidopteran species [36,37,38]. Because the CPG family does not have a highly conserved domain, we determined the CPG genes of *A. pernyi* via homology with *B. mori* CPG genes and determined whether the sequences were glycine-rich. The results indicated a great difference in the number of CPG genes between *A. pernyi* (8) and *B. mori* (22). ApCPGs had sequence identities from 45% to 91% with BmorCPGs.

In *A. pernyi*, the remaining 22 CPs were categorized into CPH (cuticular proteins hypothetical), a family with presumptive CPs in the absence of direct evidence proving them to be CPs [25]. The number of CPH genes in *A. pernyi* (22) was nearly identical to that in *B. mori* (23). Most CPHs were assigned based on the sequence similarity with known CPs, or with the tiny motif of AAP [AVL], such as GWHPABGR012725, GWHPABGR002236 and GWHPABGR012712 containing six, seven and ten repeats, respectively. Twenty-two ApCPH proteins shared more than 36% sequence identities with BmorCPG proteins (Appendix A).

### 2.4. Expression of CP Genes in the Epidermis

We matched the identified CP genes against the transcriptome data of larval middle region epidermis (not the whole epidermis) to characterize their expression patterns between *A. pernyi* (Appendix A) and *B. mori* (Appendix A). Transcriptome-based analyses indicated that 132 (60.8% of total genes) of the CP genes were expressed in the larval middle region epidermis of *A. pernyi*, a comparable number with *B. mori* where 115 (48.7% of total) were expressed (Table 1). As a whole, a highly similar expression profile was observed in the larval epidermis between two silkworm species (Appendix A).

There were 85 CPR genes (42 RR-1, 33 RR-2, 2 RR-3 and 8 RR-NC) expressed in the larval epidermis of *A. pernyi*, which accounted for 54% of the total number of CPR genes. In *B. mori,* 53 CPR genes (36 RR-1, 11 RR-2, 3 RR-3 and 3 RR-NC) were found to be expressed in the larval epidermis, comprising 36% of the total number of CPR genes (Table 1). RR-2 family genes contributed to the difference in the number of CPR family genes between *A. pernyi* and *B. mori*. FPKM analysis showed that most RR-2 genes were not expressed in the larval epidermis of *A. pernyi* and *B. mori*, and even the expressed genes had an extremely low expression level with an FPKM value ranging from 0 to 311. The numbers of highly expressed (FPKM > 1000) genes in *A. pernyi* and *B. mori* were 13 and 10, respectively. Among them, the FPKM values of ApCPR3, ApCPR34-40, BmorCPR2, BmorCPR4, BmorCPR5, BmorCPR28, BmorCPR38, BmorCPR41 and BmorCPR46 exceeded 10,000. These genes all belonged to the RR-1 family. The expressions of CPR genes in the larval epidermis of *A. pernyi* and *B. mori* were similar (Appendix A). Consistent with many other previous works [7,28,36,38], this study demonstrated that RR-1-type genes were abundant in the soft and flexible cuticles of the larval stages [33,36,39]. Although some CPR genes are highly expressed in the larval epidermis, we cannot conclude that the function of insect epidermis against harsh environments depends on these specific genes.

A total of 14 (78% of total) and 17 (74% of total) CPAP genes were expressed in the larval epidermis of *A. pernyi* and *B. mori*, respectively. For the CPAP1 subfamily, ApCPAP1-D, ApCPAP1-B2, ApCPAP1-O, ApCPAP1-L, BmorCPAP1-A, BmorCPAP1-C, BmorCPAP1-D, BmorCPAP1-N and BmorCPAP1-O were not expressed. The remaining CPAP1 genes were expressed at a low level (FPKM < 1000). However, within the CPAP3 subfamily, only BmorCPAP3-E4 had no expression. In addition, CPAP3 family genes were generally expressed at higher levels than CPAP1 family genes. The expression quantity of CPAP genes in the epidermis of *A. pernyi* and *B. mori* was similar to the proportion of high-expression genes (Appendix A), which indicated that CPAP proteins may be closely related to the cuticle function of the lepidopteran species.

The expression level of ApCPF was low (FPKM ≤ 10) and BmorCPF was not expressed in the epidermis. For the CPFL family, ApCPFL1, ApCPFL2 and BmorCPFL1 were expressed. It has been reported that the peak expression of BmorCPFL4 was before and after pupation, which might be associated with the synthesis of both exocuticle and endocuticle [40]. However, there was no expression except for CPFL1 in the epidermis of the fifth instar larva of *B. mori*, indicating that the CPFL4 gene may not play a role in the cuticle but may function through other locations and act on the cuticle. The high expression of the CPFL1 gene suggests that it may act on the cuticle. In the same way, the expression levels of the two CPFL genes in the larval epidermis of *A. pernyi* were generally low, which may have just started to be expressed in this period.

ApCPT1-4, BmorCPT1 and BmorCPT3 were expressed in the epidermis. Both ApCPT1 and BmorCPT1 genes were expressed at a high level (FPKM > 1000). It has been shown that BmorCPT1 is involved in chitin-binding and immune-related functions [41], indicating that the CPT1 gene of *A. pernyi* may also be involved in similar functions.

The expression levels of ApCPCFC2, ApCPCFC3 and BmorCPH1 were low in the larval epidermis, with FPKM values ranging from 0.2 to 10.1, and ApCPCFC1 was not expressed. This indicates that CPCFC genes may not be involved in the cuticular function of this period.

The expression levels of ApCPLCP1-2, BmorCPG7-8, BmorCPG12 and BmorCPG24 were very low in the larval epidermis, with FPKM values ranging from 0.13 to 1.03. BmorCPG13 exhibited the highest expression value, with an FPKM value of 143.57, but BmorCPG11 had no expression.

In the larval epidermis transcriptome, four (50% of total) ApCPG and fourteen (64% of total) BmorCPG genes were expressed. In *A. pernyi*, ApCPG6 exhibited the highest expression value, with an FPKM value of 1,055. In *B. mori*, BmorCPG17 showed the highest FPKM value of 5,723. ApCPG2, BmorCPG5, BmorCPG10, BmorCPG16 and BmorCPG21 were highly abundant (FPKM > 100).

The expression of CPH genes in the larval epidermis of *A. pernyi* and *B. mori* accounted for 80% (18) and 67% (15) of the total, respectively. Compared to other CP genes, CPH genes have a high-expression ratio and expression level in the larval epidermis of *B. mori*. In the larval epidermis, ApCPH13 and BmorCPH24 had the highest expression levels, with FPKM values of 646 and 10,886, respectively. BmorCPH25 and BmorCPH34 were defined as having a high expression (FPKM > 1000) in the epidermal transcriptome of *B. mori*. The results showed that CPH genes may play an important role in the larval cuticle.

### 2.5. Expression of CP Genes in the Prothoracic Gland

The transcriptome-based expression analysis mentioned above indicated that 59 (64% of the total family genes) and 73 (87%) RR-2 genes were not expressed in the larval epidermis of *A. pernyi* and *B. mori*, respectively. To address the question of where most RR-2 genes are expressed, the transcriptome data of the larval prothoracic gland available were further examined. Surprisingly, we found that 118 (54% of total genes) and 197 (83%) CP genes were expressed in the prothoracic gland of *A. pernyi* and *B. mori*, respectively (Table 1), indicating a distinct expression profile between two silkworm species (Appendix A).

There were 78 CPR genes (41 RR-1, 29 RR-2, 2 RR-3 and 6 RR-NC) and 113 CPR genes (44 RR-1, 60 RR-2, 4 RR-3 and 5 RR-NC) expressed in the *A. pernyi* and *B. mori* prothoracic gland, respectively. Like in the larval epidermis, the big difference in the number of CPR family genes expressed between *A. pernyi* and *B. mori* was also caused by the RR-2 family. However, the number of RR-2 family genes expressed in the prothoracic gland of *B. mori* (71%) was higher than that in *A. pernyi* (31.5%). The five highly expressed genes (FPKM > 1000) of *A. pernyi* belonged to the RR-1 family, while three of six highly expressed genes of *B. mori* came from the RR-1 family and the remaining three came from the RR-2 family.

### 2.6. Expression of CP Genes in Other Non-Epidermis Tissues/Organs

Besides the larval epidermis and prothoracic gland of *A. pernyi* and *B. mori*, our collection of transcriptome data also contained larval haemolymph and the midgut of *A. pernyi*, female and male adult antenna of *A. pernyi* and larval corpus allatum of *B. mori* (Table 1). Transcriptome-based expression analyses indicated that, among six tissues/organs of *A. pernyi*, the highest number of CP genes expressed was observed in the epidermis (132; 61% of the total), and the fewest CP genes expressed were found in the midgut (55; 25% of the total). Transcripts corresponding to RR-1 proteins were found mainly in the larval epidermis and prothoracic gland, but relatively fewer were found in the adult antenna. Approximately 31–42% of transcripts corresponding to RR-2 proteins were found in the larval epidermis, prothoracic gland and female and male adult antenna, while only 6.5–12% of transcripts corresponding to RR-2 proteins could be found in the larval haemolymph and midgut.

The heat maps of transcripts for each CP gene in *A. pernyi* confirm the tissue-specific expression pattern (Figure 7A), as previously reported in other insects [7,36,42]. For example, ApCPH1–2 genes were dominantly expressed in the midgut. ApCPR100-101 genes were expressed mainly in the antenna. ApCPH9-10 genes were highly expressed in the prothoracic gland. A cluster of ApCPR34-41 genes was highly expressed in the larval epidermis and prothoracic gland.

It is a surprise that, among three larval tissues/organs of *B. mori*, less than half of CP genes expressed were found in the larval epidermis (115 out of 236 CP genes), while more than 83% of CP genes expressed were observed in the prothoracic gland (197 out of 236 CP genes) and corpus allatum (203 out of 236 CP genes). The number of transcripts corresponding to RR-1 proteins were similar in the larval epidermis, prothoracic gland and corpus allatum. More than 71% (60 out of 84) of transcripts corresponding to RR-2 proteins were found in the larval prothoracic gland and corpus allatum, while only 13% (11 out of 84) of transcripts corresponding to RR-2 proteins could be found in the larval epidermis. BmorCPT3 and BmorCPH18 genes showed the highest expression levels in the corpus allatum and prothoracic gland, with FPKM values of 11,636 and 13,988, respectively. The heat map of transcripts for each CP gene in *B. mori* also confirmed the tissue-specific expression pattern (Figure 7B). For example, three BmorCPH genes (CPH24, 25 and 34) were highly expressed in the larval epidermis, but not in the prothoracic gland and corpus allatum. Although the numbers of CP genes expressed were nearly identical between the prothoracic gland and corpus allatum, the tissue-specific expression pattern was also reflected by the heat map.

## 3. Discussion

This is the first genome-wide identification of CP genes in *A. pernyi*, even for the Saturniidae species. The physical properties of the insect cuticle are varied, and most of the cuticle consists basically of CPs. To understand the molecular basis underlying the specific properties of the insect cuticle, an important initial step is to systematically identify and characterize CPs, and this knowledge will likely benefit the survival and feeding of economical insects. Previous works have indicated that CPs vary greatly in sequence and structure within and between species [7,10,24]. In the present study, we identified 217 CPs in the recently released whole genome of *A. pernyi* via sequence characterization or based on homology with known CPs of *B. mori*. All of these CP genes of *A. pernyi* were further classified into nine families: CPR, CPAP, CPF, CPFL, Tweedle, CPLCP, CPCFC, CPG and CPH. This work provided an overview of CPs in *A. pernyi*, lays a basis for further functional studies of CP genes and will help to understand CP diversity in wild silkworms.

The studied CP genes were distributed over many chromosomes in both *A. pernyi* and *B. mori*. It would be interesting to compare the position of the orthologous genes that occupy the genomes of *A. pernyi* and *B. mori*. Since *A. pernyi* has 49 chromosomes in its genome, while *B. mori* only has 28 chromosomes, this provides an opportunity to trace how chromosomal rearrangements affect the gene distribution map. The comparison, as shown in Appendix A, indicated that the position of CP orthologous gene clusters is relatively stable, suggesting that chromosomal rearrangement did not affect the CP gene distribution map. So far, CP genes have been identified at the genome level in six lepidopteran insects including *A. pernyi*. Among these lepidopteran insects, the numbers of CP genes vary: 158 in *Danaus plexippus* L. [12,30], 197 in *Dendrolimus punctatus* Walker [38], 217 in *A. pernyi*, 236 in *B. mori*, 246 in *Manduca sexta* L. [28] and 287 in *Spodoptera litura* Fabricius [36]. The comparison of the number of CPs among six lepidopteran insects confirmed that the numbers of CPs are diverse [7,9,10]. According to transcriptome and genomic data, the varied number of CP genes has also been identified in many insect species of non-Lepidoptera order, for example, *B. dorsalis* (164) [7], *A. sinensis* (250) [33], *A. gambiae* (268) [8,20,24] and *Drosophila melanogaster* Meigen (174) [43], further supporting the fact that the numbers of CPs are diverse among insect species. Our work revealed a comparable number of CP genes between *A. pernyi* and *B. mori*, with both of them belonging to the superfamily Bombycoidea. A further comparison indicated that the numbers of CPLCP and CPG families mainly contributed to the number variation in CP genes between the two Bombycoidea species, which may have resulted in the softer larval epidermis in *B. mori* compared to that in *A. pernyi*. Recent work into *B. mori* has indicated that a highly expressed CPLCP gene in the larval epidermis is able to regulate the larval body shape [44].

The present study found that *A. pernyi* and *B. mori* possess similar numbers of CPR genes; although, species-specific duplication events have occurred in the RR-2-type CPs. The larvae of two silkworm species have a relatively large size and spend their lives in the open field; so, they have developed various exoskeletal shapes, such as spines and tubercles. The similar gene numbers of CPR proteins between *A. pernyi* and *B. mori* may allow for the possibility of forming a similar body surface and scale structures to adapt to the environment. Thus, this can be explained by adaptive evolution [7]. Moreover, many RR-2-type CPRs formed several monophyletic groups according to the taxa in the phylogenetic tree, suggesting gene tandem duplications where the sequence clusters have evolved from an ancestral gene and expanded separately after speciation [7,25,28]. It has been postulated that this species-specific duplication of CPs is associated with taxa-specific exoskeletal characteristics [7]. A recent study has also suggested that gene duplication and tandem duplication may be correlated with the living environment of the lepidopteran species and their epidermal defense mechanism [38], which indicates that the RR-2 family plays an important role in resisting the harsh external environment of *A. pernyi*.

Our work provided the functional cue for the RR-1 and RR-2 proteins of the CPR family. It has been reported that RR-1 proteins are prevalent in the soft cuticles and RR-2 proteins are often associated with the hard cuticle [7,18,39]. In agreement with this, we found similar number of RR-1 genes expressed in the larval epidermis and prothoracic gland between *A. pernyi* and *B. mori*, which suggests that their hardness would be determined by the number of RR-2 genes expressed. In the larval epidermis, *A. pernyi* (33) had more RR-2 genes expressed than *B. mori* (11), thus indicating a harder epidermis for *A. pernyi* and a softer epidermis for *B. mori*. As for the prothoracic gland, *A. pernyi* (29) had less RR-2 genes expressed than *B. mori* (60), thus indicating a softer prothoracic gland for *A. pernyi* and a harder prothoracic gland for *B. mori*.

Our study provided the transcriptome-based expression profiles of CP genes in the larval corpus allatum, prothoracic gland and haemolymph. The expressions of CP genes in the corpus allatum and prothoracic gland were observed in the larval stage of *B. mori* where the corpus allatum and prothoracic gland are separate organs. Generally, previous studies focused on CPs by investigating the transcription profiles in the larval epidermis as well as the non-epidermis tissues/organs [7,25,28,45]. So far, in insects, CP genes have been found to be expressed in a large number of non-epidermis tissues/organs, such as the nervous system, antenna, fat body, ovary or testis, brain, silk gland, male accessory gland, midgut and muscle [7,10,25,28]. However, one should be aware that contamination of tracheal cells during the dissection of the tissues might result in false positive results [7]. The trachea can penetrate most of the tissues, including the prothoracic gland and corpora allatum, which also might result in false positive results. Thus, we have to state that the possibility of the detection of CP gene expression from the trachea in non-epidermal tissues was not ruled out. Our work provided evidence for the expression of CP genes in the haemolymph of *A. pernyi*, where the expression of CP genes is interesting. The CPs synthesized in the non-epidermis cells perhaps make use of them in these organs [46]. This is supported by a study on *B. mori*, where a CPT1 gene was detected in the fat body and was proven to participate in silkworm innate immunity via the recognition of *Escherichia coli* [41].

Note that we used the larval middle region epidermis in the fifth instar as the “larval epidermis” and found that 60.8% of CPR genes were expressed in the larval epidermis of *A. pernyi*. This case might also be observed in the larval epidermis of *B. mori* where 48.7% of CP genes were expressed [47]. This would lead to the underestimate of the number of CP genes expressed in the larval epidermis. A previous study has shown that larvae have other types of cuticles on the head capsule, claw and proleg [28]. Thus, other CP genes would be expressed in other regions of the larval cuticle. Since CP distribution is highly diverse not only among body regions but also in developmental stages, the temporal and spatial expression profiles of CP genes in the whole epidermis of a certain insect species need to be further addressed in the future.

## 4. Materials and Methods

### 4.1. Identification of Cuticular Protein Genes

The chromosome-scale genome of *A. pernyi* [31] was obtained from Genome Warehouse (GWH) in the National Genomics Data Center (NGDC), Beijing Institute of Genomics (BIG), Chinese Academy of Sciences, under accession numbers CRA002120 (https://bigd.big.ac.cn/gsa/, accessed on 20 December 2021) and GWHABGR00000000 (https://bigd.big.ac.cn/gwh/, accessed on 20 December 2021), respectively.

The CPR family is recognized by an extended version of the R&R Consensus, chitin_bind_4 (PF00379) [48,49]; so, we searched the whole genome sequence with PF00379 using HMMER v3.0 (http://hmmer.janelia.org/, accessed on 10 February 2022) [50] based on profile hidden Markov models (pHMMs) [51]. We then classified the members of the CPR family into subfamilies using the CuticleDB website (http://bioinformatics.biol.uoa.gr/cuticleDB/, accessed on 25 March 2022) and BLASTp search [52]. We identified the CPT, CPF, CPAP and CPLCA families using the Tweedle motif (PF0310), 44 aa consensus (PF11018), retinin domain (PF04527) and ChtBD2 domain (PF01607), respectively. The remaining CPFL, CPLCP, CPCFC, 18 aa, CPG and CPH families were identified using the verified CP sequences via BLASTp. Sequence logos were generated from the CPs of *A. pernyi* and *B. mori* using WEBLOGO (http://weblogo.berkeley.edu/logo.cgi, accessed on 13 June 2022) [34]. All of the identified genes based on sequence similarity were also checked manually using the diagnostic features. We calculated the sequence similarity using TBtools v 1.099 [53]. Finally, the presence of a predicted signal peptide in the presumed CPs was checked using the SignalP4.1 [54] online program (https://services.healthtech.dtu.dk/service.php?SignalP-4.1, accessed on 17 June 2022). According to the nomenclature used by *D. melanogaster*, *B. mori* and *A. gambia*, these identified CP genes have been named corresponding to the type of CPs.

### 4.2. Phylogenetic Classification

Phylogenetic classification was performed on CP sequences from *B. mori* together with our sequences. We aligned the full amino acid sequences of CP genes using the MAFFT v7 [55] online service. The amino acid sequences (CPRs, CPAPs and CPTs) in *A. pernyi* and *B. mori* were performed to analyze the phylogenetic relationships. For the CPR group, the extended R&R Consensus was used; RR-1 and RR-2 subgroups were treated separately. Phylogenetic trees were constructed using MEGA X (v10.2.4) [56] and the neighbor-joining (NJ) method with a Poisson correction model. Gaps were treated using the pairwise deletion method and statistical analysis was performed via the bootstrap method using 1000 repetitions. Phylogenetic trees were visualized and edited using the iTOL v5 online service (https://itol.embl.de/, accessed on 5 July 2022) [57]. The chromosomal locations of the CP genes of *A. pernyi* and *B. mori* were mapped using MG2C_V2.1 (http://mg2c.iask.in/mg2c_v2.1, accessed on 7 September 2022) [58].

### 4.3. Transcriptome-Based Expression Pattern

The *A. pernyi* strain Qing 6 exhibiting a yellow-cyan skin color was used for transcriptome sequencing. The fifth instar larvae on Day 5 were used to isolate the epidermis, midgut and haemolymph. The haemolymph was first collected and immediately frozen in liquid nitrogen. The larval middle region epidermis (not the whole epidermis) was dissected on ice with the removal of the attached muscles and fat bodies. The isolated epidermis and midgut were immediately frozen in liquid nitrogen. Transcriptome sequencing was completed by Biomarker Technologies (Beijing, China) using the Illumina HiSeq 2500 platform (CA, USA). These transcriptome data were deposited in the SRA database in NCBI with accession numbers SRX18549155 for the epidermis, SRX18549156 for the haemolymph and SRX18549157 for the midgut, respectively (Appendix A). Three pieces of transcriptome data of the male antennae, female antennae and prothoracic gland of *A. pernyi* were also download from the SRA database with accession numbers SRR21694239, SRR21694240 and SRR5119598, respectively. Trimmomatic was used to process the raw data to remove low-quality reads [59]. The clean reads of six transcriptome data of *A. pernyi* were mapped to the reference genome sequence (GWHABGR00000000, https://bigd.big.ac.cn/gwh/, accessed on 20 December 2021) [31]. Hisat2 tools (v2.1.0) [60] was used to map with the reference genome.

For *B. mori*, three transcriptome data from the larval epidermis [47], corpus allatum [61] and prothoracic gland [62] were downloaded from the SRA database with accession numbers DRR077426, SRR12830833 and SRR14306779, respectively. The clean reads were mapped to the reference genome sequence (https://ftp.ncbi.nlm.nih.gov/genomes/all/GCF/014/905/235/GCF_014905235.1_Bmori_2016v1.0/, accessed on 28 December 2021) [32].

For the gene expression analysis, the number of expressed tags was calculated and then converted to FPKM (fragments per kilobase per million) using Stringtie (v2.2.1) [60]. The expression level of the identified CP gene was calculated based on the FPKM values from six pieces of transcriptome data of *A. pernyi* and three pieces of transcriptome data of *B. mori.* The FPKM values ranged from 0 to greater than 10,000 and gene expression was grouped as follows: a value greater than 0 and less than 1 indicates no or extremely low expression; 1–10, very low expression; 10–100, low expression; 100–1000, moderate expression; 1000–10,000, high expression; greater than 10,000, very high expression. The log10 transformation of FPKMs was used to construct the heat maps using the Pheatmap package [63] in R programming [64].

## 5. Conclusions

This study, for the first time, identified CPs in the genome of *A. pernyi* and investigated the expression profile in the epidermis as well as several non-epidermis tissues/organs of *A. pernyi* and *B. mori*. We identified 217 CPs (157 CPR, 4 Tweedle, 1 CPF, 2 CPFL, 3 CPCFC, 12 CPAP1, 6 CPAP3, 8 CPG and 22 CPH) in the genome of *A. pernyi*, a comparable number to *B. mori* (236 CPs). We found that the CPLCP and CPG families were the main contributors to the number difference in CP genes between two silkworm species. The analysis of a transcriptome-based expression pattern suggested that the hardness difference in the epidermis and prothoracic gland between species may be caused by the number of RR-2 genes expressed. Transcriptome analysis of the three tissues/organs of *B. mori* revealed that the numbers of CP genes expressed in the corpus allatum and prothoracic gland were higher than that in the epidermis. Our work provided an overall framework for functional research into the CP genes of Saturniidae.

## Figures and Tables

**Figure 1 ijms-24-06991-f001:**
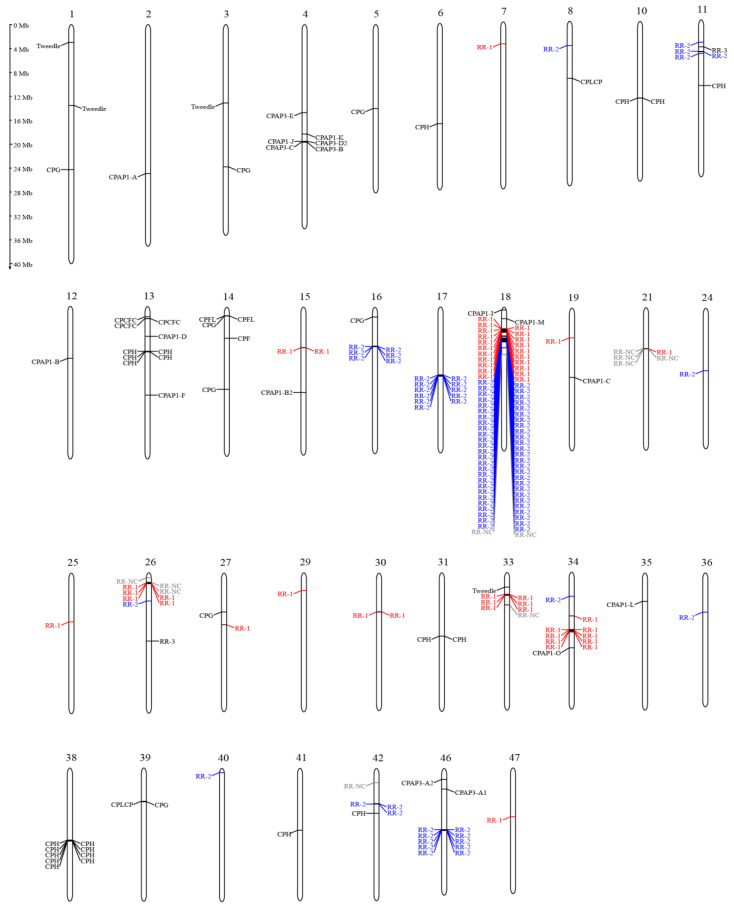
Chromosomal localization of cuticular protein genes in *A. pernyi*. (For details of the cuticular protein genes in this figure, the reader is referred to Appendix A.)

**Figure 2 ijms-24-06991-f002:**
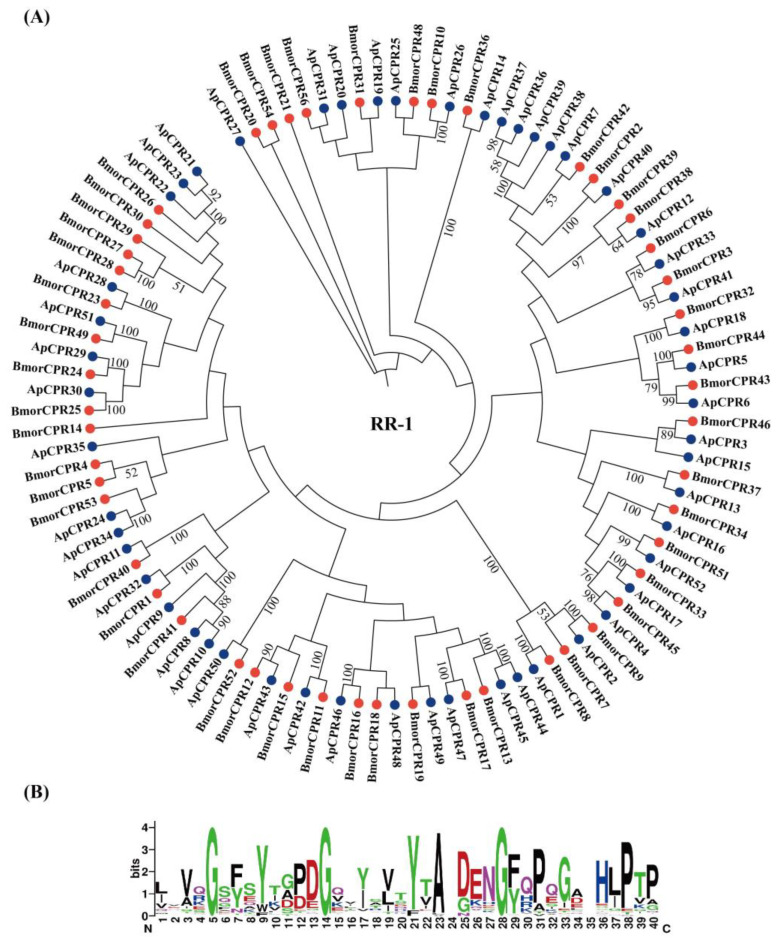
Phylogenetic relationship (**A**) and sequence logo of the conserved region (**B**) of RR-1 proteins from *A. pernyi* (Ap; blue circle) and *B. mori* (Bmor; red circle). The sequence logo was generated by Weblogo [34].

**Figure 3 ijms-24-06991-f003:**
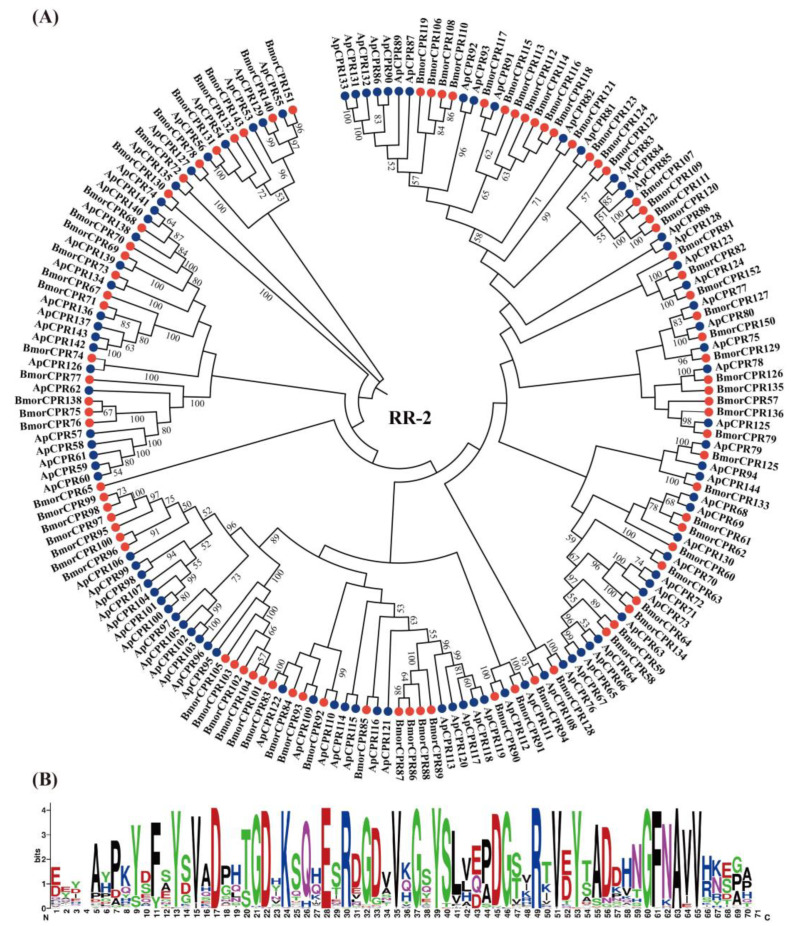
Phylogenetic relationship (**A**) and sequence logo of the conserved region (**B**) of RR-2 proteins from *A. pernyi* (Ap; blue circle) and *B. mori* (Bmor; red circle). The sequence logo was generated by Weblogo [34].

**Figure 4 ijms-24-06991-f004:**
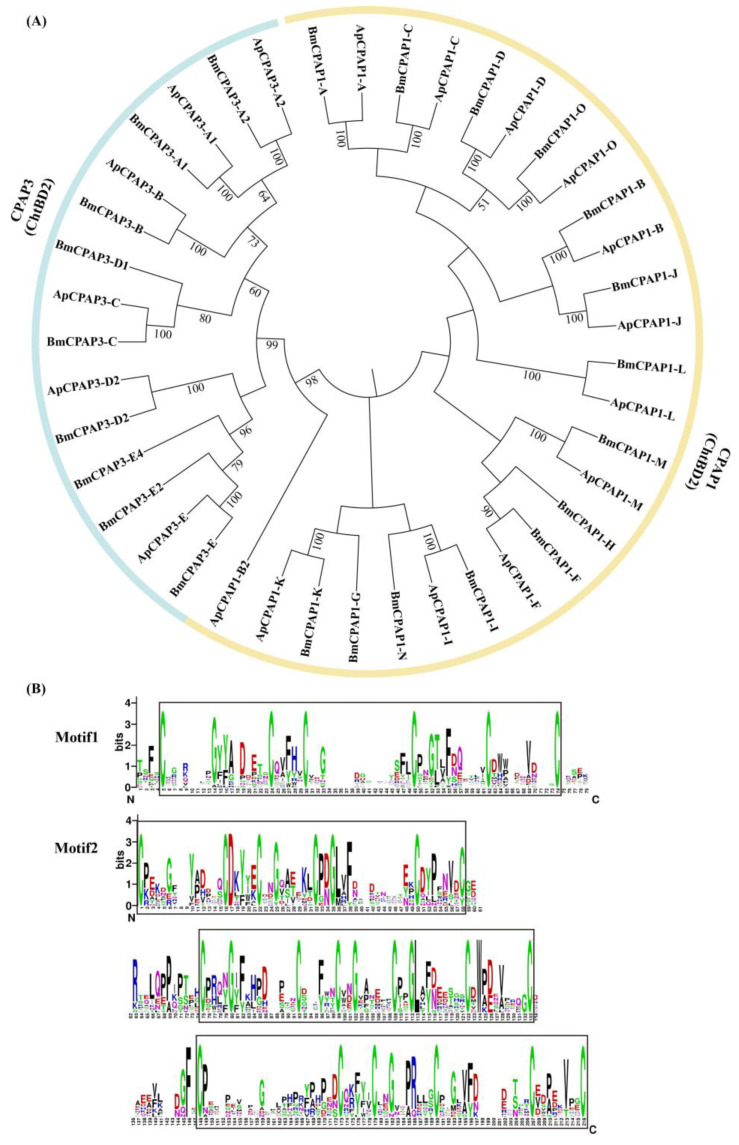
Phylogenetic relationship (**A**) and domain organization (**B**) of CPAP1 and CPAP3 proteins in *A. pernyi* (Ap) and *B. mori* (Bmor). The pink clades represent the ChtBD2s for CPAP1 proteins; the purple clades represent the ChtBD2s for CPAP3 proteins. The sequence logo shows the conserved region of the CPAP proteins in *A. pernyi*. The protein with one Cx_14–16_Cx_5_Cx_9–13_Cx_12_Cx_7–8_C motif is CPAP1 and the one with three Cx_13–24_Cx_5_Cx_9–10_Cx_12–16_Cx_7–8_C motifs is CPAP3. The chitin-binding domains (ChtBD2) are represented by squares. The sequence logos were generated by Weblogo [34].

**Figure 5 ijms-24-06991-f005:**
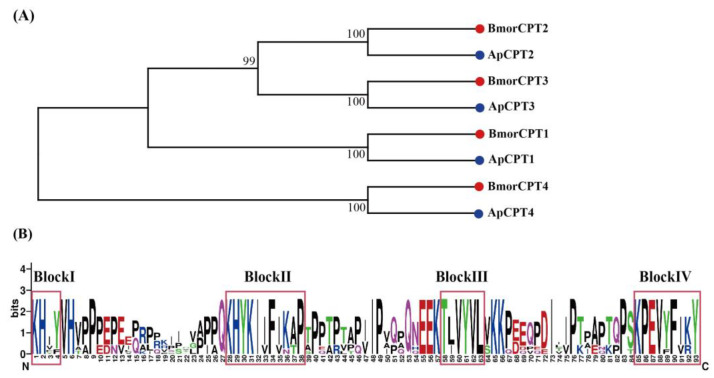
Phylogenetic tree (**A**) and sequence logo of the conserved region (**B**) of Tweedle protein family in *A. pernyi* (Ap; blue circle) and *B. mori* (Bmor; red circle). The sequence logo was generated by Weblogo [34].

**Figure 6 ijms-24-06991-f006:**
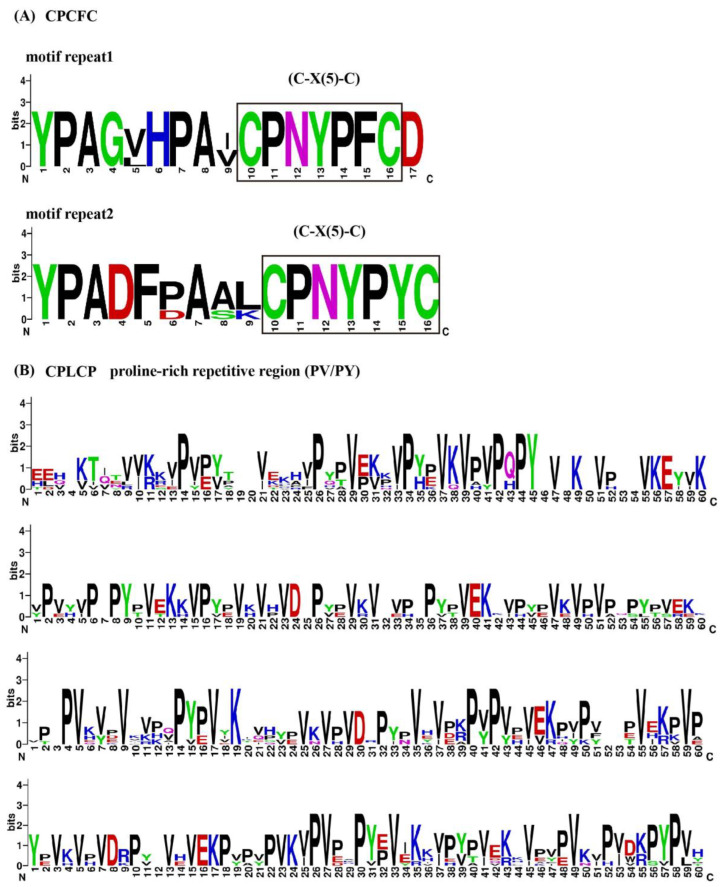
WebLogos of the CPCFC (**A**) and CPLCP (**B**) in *A. pernyi*. CPCFC sequences have two repeats of a motif ending in C-X_5_-C, and CPLCP sequences have the presence of a high density of PV and PY. The sequence logos were generated by Weblogo [34].

**Figure 7 ijms-24-06991-f007:**
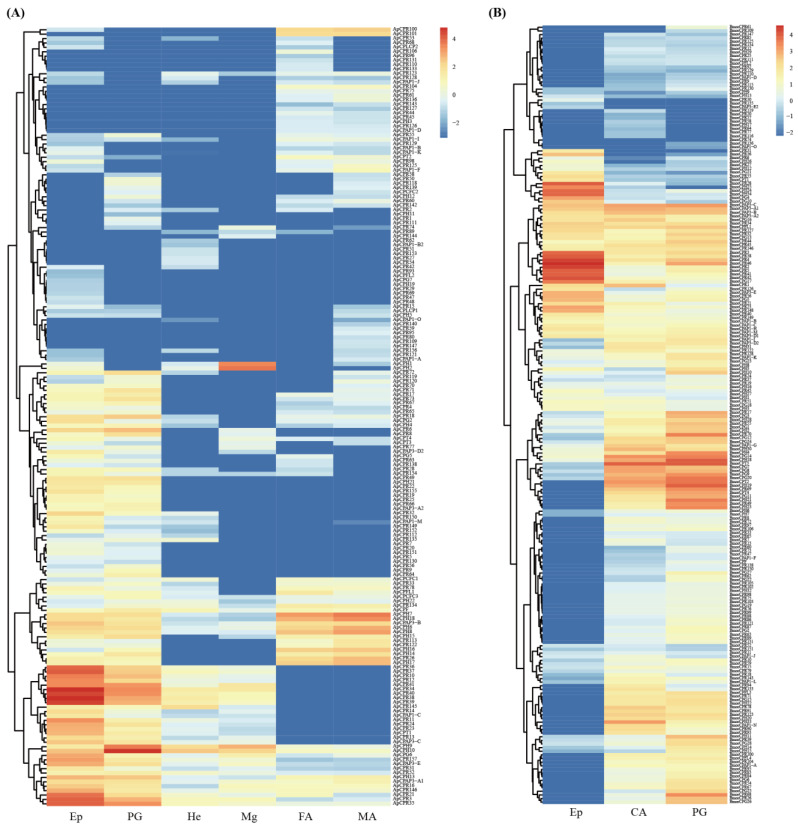
The expression profiles of CP genes in six tissues/organs of *A. pernyi* (**A**) and three tissues/organs of *B. mori* (**B**). Red indicates a high expression and blue indicates a low expression. The analyzed tissues/organs include the epidermis (Ep), prothoracic gland (PG), Haemolymph (He), midgut (Mg), female antenna (FA) and male antenna (MA).

**Table 1 ijms-24-06991-t001:** The number of cuticular proteins identified in the genome and expressed in the epidermis and non-epidermis tissues/organs of *A. pernyi* and *B. mori*. A total of 217 CP genes were identified in the genome of *A. pernyi*, and 236 CP genes were retrieved in the latest genome assembly of *B. mori*. The number of CP genes expressed was calculated based on the transcriptomic data. Ep, epidermis; PG, prothoracic gland; He, haemolymph; Mg, midgut; FA, female antenna; MA, male antenna; CA, corpus allatum.

Family		*A. pernyi*							*B. mori*		
	Genome	Ep	PG	He	Mg	FA	MA	Genome	Ep	PG	CA
CPR_RR-1	52	42	41	28	22	15	15	51	36	44	44
CPR_RR-2	92	33	29	11	6	34	39	84	11	60	63
CPR_RR-3	2	2	2	2	2	1	1	4	3	4	4
CPR_RR-NC	11	8	6	7	2	2	3	6	3	5	5
CPAP1	12	8	4	7	1	6	9	14	9	14	13
CPAP3	6	6	6	4	5	3	3	9	8	7	7
CPCFC	3	2	3	2	1	2	3	1	1	1	1
CPLCP	2	2	1	0	0	1	1	6	5	6	6
CPLCA	0	0	0	0	0	0	0	2	2	2	2
CPF	1	1	1	1	1	1	1	1	0	1	1
CPFL	2	2	1	1	0	1	1	4	1	4	4
CPT	4	4	4	1	3	2	3	4	2	4	4
CPG	8	4	3	2	1	3	2	22	14	22	22
CPH	22	18	17	11	11	15	17	23	15	21	22
18aa	0	0	0	0	0	0	0	5	5	2	5
Total	217	132	118	77	55	86	98	236	115	197	203

## Data Availability

Data are contained within the article or Appendix A.

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
