# Peer review of "Genome-Wide Identification and Transcriptome-Based Expression Profile of Cuticular Protein Genes in Antheraea pernyi"

_ijms, 2023, doi:10.3390/ijms24086991_

Round 1

Reviewer 1 Report

In the present study, Fu et al. performed the genome wide identification and expression analysis of cuticle protein genes in Antheraea pernyi. Antheraea pernyi is an insect of economic significance as a silk producer and an edible resource. The authors identified the numbers, sequences, structures, and expression patterns of cuticle protein genes in this species. Given the importance of cuticle proteins in body shaping and stress resistance, this work is interesting and valuable for related researchers in the field of Antheraea pernyi study. The results and conclusions in this manuscript provide significant advances in this gene family and may be useful for further studies on molecular mechanisms related to cuticle proteins in this insect. Overall, the manuscript is well written and the experiments are correctly conducted, however, there are still a few points that need to be improved before it is acceptable for this journal.

1. In Figure 1, the chromosome numbers are incompletely displayed.

2. There are too many figures in the manuscript and some of the figures did not provide enough information. The authors are suggested to combine some of the related figures into one. For example, figure 7 and figure 8.

Author Response

Point 1: In Figure 1, the chromosome numbers are incompletely displayed.

Response 1: Addressed. We were really sorry for our careless mistakes. The picture has been modified.

Point 2: There are too many figures in the manuscript and some of the figures did not provide enough information. The authors are suggested to combine some of the related figures into one. For example, figure 7 and figure 8.

Response 2: Addressed. Thanks for your suggestion. We have merged the pictures 7 and 8.

Reviewer 2 Report

This is a very thorough characterization of cuticle proteins identified in the recently released genome sequence of the silk worm, A. pernyi.   Absence of certain gene families, 18aa and CPLPA, are noted, as well as species specific gene duplications and losses.   In general the conclusions that cuticle hardness is correlated with specific CP gene expression appears to be well founded.  There are a few issues noted below that would clarify some of the arguments made.

The use of homolog and ortholog are generally clear. However, on line139, I think ortholog is a better choice, since the RR-2 genes found would be homologs, while not direct orthologs.

Similarly on line 186.  1-1 orthologs seems a clearer choice.

In line 253, it is not clear what the authors mean by ”the exercise of insect epidermal function” do they mean the function of insect epidermis?

In line 400, is there a true causal connection?  Would correlated be more correct? Also the grammar is not correct in line 401, should the period be a comma?

Line 506, the grammar is a little confused, do they mean the genes expressed were outnumbered

Author Response

Point 1: The use of homolog and ortholog are generally clear. However, on line139, I think ortholog is a better choice, since the RR-2 genes found would be homologs, while not direct orthologs.

Response 1: Addressed.

This sentence has been modified to “The remaining RR-2 genes have no clear orthologous genes.”.

Point 2: Similarly on line 186. 1-1 orthologs seems a clearer choice.

Response 2: Addressed.

This sentence has been modified to “ApCPF and BmorCPF were 1-1 orthologous genes with high sequence identity of 81%.”.

Point 3: In line 253, it is not clear what the authors mean by “the exercise of insect epidermal function” do they mean the function of insect epidermis?

Response 3: Addressed.

This sentence has been modified to “Although some CPR genes are highly expressed in the larval epidermis, we cannot conclude that the function of insect epidermis against harsh environments depends on these specific genes.”.

Point 4: In line 400, is there a true causal connection? Would correlated be more correct? Also the grammar is not correct in line 401, should the period be a comma?

Response 4: Addressed.

This sentence has been modified to “A recent study has also suggested that gene duplication and tandem duplication may be correlated with the living environment of lepidopteran species and their epidermal defense mechanism [38], which indicates that RR-2 family plays an important role in resisting the harsh external environment of A. pernyi.“.

Point 5: Line 506, the grammar is a little confused, do they mean the genes expressed were outnumbered.

Response 5: Addressed.

This sentence has been modified to “Transcriptome analysis of the three tissue/organs of B. mori revealed that the numbers of CP genes expressed in the corpus allatum and prothoracic gland are higher than that in the epidermis.”.

Reviewer 3 Report

     Insect cuticle is an extra cellular matrix covering insect body and functions as exoskeleton. The mechanical property of cuticle varies, and this feature supports variable insect life. The mechanical property of cuticle depends on the component of cuticle, especially cuticular proteins (CPs), and a degree of sclerotization, cross-linking between CPs or CP and chitin fiber. Therefore, to understand the mechanism of construction of diverse cuticle in each species, it is important to identify CPs and to reveal their distribution in the species. Authors identified all CPs from Antheraea pernyi using genome and transcriptome data. They also analyzed tissue distribution of CPs by use of transcription data and compared the results with another well-studied lepidopteran insects Bombyx mori. Although the identification and structural analysis of CP genes are acceptable after some modifications, there are some critical problems in the expression analyses as follows. 

1. CP gene expression in “epidermis”

Authors used larval epidermis as the sample for “epidermis”. By comparing expression data from this sample and other tissues, they concluded that many CP genes are not expressed in epidermis. They did not specify the region of the larval epidermis. If it was a small area of larval epidermis, the expression data represent only a specific type of cuticle. At least B. mori data for epidermis that they used for comparison was from a specific area of larval epidermis. Furthermore, CP distribution are highly diverse among not only body regions but also developmental stages. Using only larval epidermis revealed CP distribution to epidermis very partially. 

2. CP gene expression in non-epidermal tissues

Trachea has cuticle on its lumen and penetrate most of the tissues. The detected expression of CP genes from non-epidermal tissues might be expression in trachea. This issue must be examined carefully. 

For the structural analyses, the points below should be also addressed.

1. Phylogenetic analyses

The protein regions used for phylogenetic analyses are not mentioned. It is not clear whether the R&R Consensus was used, or full protein region was used for CPRs. Which chitin binding domain was used for CPAP3?

2. the extended R&R Consensus and pfam00379

The extended R&R Consensus and pfam00379 are not the same exactly. Authors used these two terms as the same. The WebLogo for RR-1 (Fig. 2B) is shorter than the extended R&R Consensus region. Because authors discussed about CPR, the extended R&R Consensus should be used rather than pfam00379. 

Author Response

Point 1: Authors used larval epidermis as the sample for “epidermis”. By comparing expression data from this sample and other tissues, they concluded that many CP genes are not expressed in epidermis. They did not specify the region of the larval epidermis. If it was a small area of larval epidermis, the expression data represent only a specific type of cuticle. At least B. mori data for epidermis that they used for comparison was from a specific area of larval epidermis. Furthermore, CP distribution are highly diverse among not only body regions but also developmental stages. Using only larval epidermis revealed CP distribution to epidermis very partially.

Response 1: In our study, the larval middle region epidermis/integument in the fifth instar at day 5 was used. The related sentences have been revised accordingly. The information that CP distribution are highly diverse among not only body regions but also developmental stages has also been added in the text.

Point 2: Trachea has cuticle on its lumen and penetrate most of the tissues. The detected expression of CP genes from non-epidermal tissues might be expression in trachea. This issue must be examined carefully.

Response 2: Revised. Please see the final paragraph in the Discussion section.

Our study provided the transcriptome-based expression profiles of CP genes in the corpus allatum and prothoracic gland where the numbers of CP genes expressed are higher than that in the larval epidermis. This case was observed in the larval stage of B. mori where corpus allatum and prothoracic gland all are separated organs. Generally, previous studies focused on CPs by investigating the transcription profiles in the larval epidermis as well as the non-epidermis tissues/organs [7, 25, 28, 44]. So far, in insects, CP genes have been found to be expressed in a large number of non-epidermis tissues/organs, such as trachea, nervous system, antenna, fat body, ovary or testis, brain, silk gland, male accessory gland, midgut, muscle [7, 10, 25, 28]. Our work also provided evidence for the expression of CP genes in the haemolymph. However, one should be aware that contamination of tracheal cells during the dissection of the tissues might result in false positive results [7]. Trachea can penetrate most of the tissues, which also might result in false positive results. Unlike A. pernyi where the prothoracic gland shows a loose structure and may have tracheal cell comtamination, B. mori corpus allatum and prothoracic gland displays intact and smooth structure and may have no tracheal cell comtamination. These CPs synthesized in the non-epidermis cells perhaps make use of them in these organs [45]. This is supported by a study in B. mori, where a CPT1 gene is detected in the fat body and has proved to participate in the silkworm innate immunity by recognition of Escherichia coli [41]. Since CP distribution are highly diverse among not only body regions but also developmental stages, the whole expression profile of CP genes in the epidermis of a certain insect species needs to be further addressed in the future.

Point 3: The protein regions used for phylogenetic analyses are not mentioned. It is not clear whether the R&R Consensus was used, or full protein region was used for CPRs. Which chitin binding domain was used for CPAP3?

Response 3: Addressed.

For phylogenetic analyses, the whole protein sequence was used in this study. The corresponding sentence has been modified.

Point 4: The extended R&R Consensus and pfam00379 are not the same exactly. Authors used these two terms as the same. The WebLogo for RR-1 (Fig. 2B) is shorter than the extended R&R Consensus region. Because authors discussed about CPR, the extended R&R Consensus should be used rather than pfam00379.

Response 4: Addressed.

The related sentences have been modified as follows: “WebLogo analysis [34] confirmed that RR-1 proteins in A. pernyi contain the conserved R & R Consensus region (GxFxYxxPDGxxxxVxYxADENGYQPxGAHLP) (Figure 2B).” and “Sequence logos generated from 176 RR-2 genes of A. pernyi and B. mori confirmed that RR-2 proteins in A. pernyi contain the conserved R & R Consensus (EYDAxPxYxFxYxDxHTGDxKSQxExRDGDVVxGxYSLxExDGxxRTVxYTADxxNGFNAVVxxE) (Figure 3B).”

Reviewer 4 Report

The work is based on genome-wide data for Antheraea pernyi and Bombyx mori available in genomic databases (at least for B. mori) and on own authors’ data on Antheraea pernyi transcriptomes. It represents the results of a comparative analysis of these two species, aimed at identifying the genes of structural cuticular proteins. The work is a solid contribution to the comparative and functional genomics in general and to Lepidoptera studies in particular and will serve as a basis for further research on the genetics and genomics of cuticular proteins.

Having said this, I note that the work is descriptive. This is not a disadvantage, but nevertheless, the work would greatly benefit if the authors made an attempt to understand how differences between species arose and how these differences are expressed in the context of mapping the studied genes and their orthologs on the chromosomes of both species. As follows from the presented Figure, the studied gene families are distributed over many chromosomes. At the very least, it would be interesting to compare them with the position the same genes occupy in the Bombyx mori genome. Since Antheraea pernyi has 47 chromosomes in its genome, while Bombyx mori has only 28 chromosomes, the authors have a unique opportunity to trace how chromosomal rearrangements affected the gene distribution map.

In the GenBank, I did not find any information on the complete genome of Antheraea pernyi, especially on chromosome assemblies (There is only data on mt genome).

The paper talks about 4 species of Lepidoptera and some Diptera. It remains unclear whether there are data on other  taxa. Information about this would be extremely important in order to fit the study into a more general context and attract more attention from readers.

On the whole, my remarks are rather advisory in nature, and are aimed at strengthening the comparative nature of the work, which is otherwise descriptive.

Author Response

Point 1: Having said this, I note that the work is descriptive. This is not a disadvantage, but nevertheless, the work would greatly benefit if the authors made an attempt to understand how differences between species arose and how these differences are expressed in the context of mapping the studied genes and their orthologs on the chromosomes of both species. As follows from the presented Figure, the studied gene families are distributed over many chromosomes. At the very least, it would be interesting to compare them with the position the same genes occupy in the Bombyx mori genome. Since Antheraea pernyi has 47 chromosomes in its genome, while Bombyx mori has only 28 chromosomes, the authors have a unique opportunity to trace how chromosomal rearrangements affected the gene distribution map.

Response 1: Addressed. A new paragraph was added in the discussion section as follow:

The studied CP genes are distributed over many chromosomes in both A. pernyi and B. mori. It would be interesting to compare the position of the orthologous genes that occupy in the genomes of A. pernyi and B. mori. Since A. pernyi has 49 chromosomes in its genome, while B. mori has only 28 chromosomes, this provides an opportunity to trace how chromosomal rearrangements affected the gene distribution map. The comparison, as shown in Table S5, indicated that the position of CP orthologous gene clusters is relatively stable, suggesting that chromosomal rearrangement did not affect CP gene distribution map.

Point 2: In the GenBank, I did not find any information on the complete genome of Antheraea pernyi, especially on chromosome assemblies (There is only data on mt genome).

Response 2: This genomic information of A. pernyi was uploaded to Genome Warehouse in National Genomics Data Center, Beijing Institute of Genomics (BIG), Chinese Academy of Sciences, under accession numbers CRA002120 (https://bigd.big.ac.cn/gsa/) and GWHABGR00000000 (https://bigd.big.ac.cn/gwh/). Please see 4.1.

Point 3: The paper talks about 4 species of Lepidoptera and some Diptera. It remains unclear whether there are data on other taxa. Information about this would be extremely important in order to fit the study into a more general context and attract more attention from readers.

Response 3: Addressed.

The related sentences have been modified to “So far, CP genes have been identified at the genome level from six lepidopteran insects including A. pernyi. Among these lepidopteran insects, the numbers of CP genes vary: 158 in Danaus plexippus L. [12, 30], 197 in Dendrolimus punctatus Walker [38], 217 in A. pernyi, 236 in B. mori, 246 in Manduca sexta L. [28] and 287 in Spodoptera litura Fabricius [36]. The comparison of the number of CPs among six lepidopteran insects confirmed that the numbers of CPs are diverse [7, 9, 10]. According to transcriptome or genomic data, the varied number of CP genes have also been identified from many insect species of non-Lepidoptera order, for example, B. dorsalis (164) [7], Anopheles sinensis Wiedemann (250) [33], Anopheles gambiae Giles (268) [8, 20, 24], Drosophila melanogaster Meigen (174) [50], further supported that the numbers of CPs are diverse among insect species.”.

Point 4: On the whole, my remarks are rather advisory in nature, and are aimed at strengthening the comparative nature of the work, which is otherwise descriptive.

Response 4: Thank you for your suggestions.

Round 2

Reviewer 3 Report

This reviewer strongly recommends stating clearly next two points for expression analysis. 

1. Larval epidermis

Authors used the preparations from the larval middle region epidermis/integument in the fifth instar as “larval epidermis” and found only 85 CPR genes were expressed in the larval epidermis. Larvae have other types of cuticle on head capsule, claws, and so on. That description may mislead readers. Authors should state clearly that they used body trunk region and other regions of larval cuticle probably have other CPRs in results and discussion sections. 

2. Preparation of non-epidermis tissues

Authors claims that prothoracic glands and corpora allata from B. mori may not include tracheae. According to this reviewer’s experiences, removing tracheae completely from tissues including prothoracic glands and corpora allata of B. mori is impossible by normal dissection technique. If authors used a special technique to remove tracheae completely or they examined the detected expression was not from tracheae but truly from non-epidermal cells, they should explain about that. If not, they should state that the possibility of detection of CP gene expression from trachea in the non-epidermal tissues was not ruled out. Expression in hemolymph might be OK, because cells in hemolymph are isolated. The expression of CP genes in hemolymph is interesting. 

Other points to be fixed are below:

1. Line 451, trachea as a non-epidermis tissue.

“The tracheae are invaginations of the epidermis and thus their lining is continuous with the body cuticle” (cited from P. J. Gullan and P. S. Cranston, The Insect: an outline of entomology, Wiley Blackwell). There is also similar explanation in another textbook (Marc J. Klowden, Physiological Systems in Insects, Academic Press). It is not appropriate to treat trachea as a non-epidermis tissue.  

2. Line 18, two silkworms

“Two silkworms” sounds like two individuals of silkworm. This should be like “two silkworm species”. The same expression is seen at line 250. 

Author Response

Point 1: Larval epidermis. Authors used the preparations from the larval middle region epidermis/integument in the fifth instar as “larval epidermis” and found only 85 CPR genes were expressed in the larval epidermis. Larvae have other types of cuticle on head capsule, claws, and so on. That description may mislead readers. Authors should state clearly that they used body trunk region and other regions of larval cuticle probably have other CPRs in results and discussion sections.

Response 1: Addressed. Thanks for your critical suggestion. Please see Line 450-459.

Note that we used the larval middle region epidermis in the fifth instar as “larval epidermis” and found 60.8% CPR genes were expressed in the larval epidermis of A. pernyi. This case might also be observed in the larval epidermis of B. mori where 48.7% CP genes were expressed [60]. This would lead to the underestimate of the number of CP genes expressed in the larval epidermis. Previous study has shown that larvae have other types of cuticle on head capsule, claw and proleg [28]. Thus, other CP genes would be expressed in other regions of larval cuticle. Since CP distribution are highly diverse among not only body regions but also developmental stages, temporal and spatial expression profile of CP genes in the whole epidermis of a certain insect species needs to be further addressed in the future.

Point 2: Preparation of non-epidermis tissues. Authors claims that prothoracic glands and corpora allata from B. mori may not include tracheae. According to this reviewer’s experiences, removing tracheae completely from tissues including prothoracic glands and corpora allata of B. mori is impossible by normal dissection technique. If authors used a special technique to remove tracheae completely or they examined the detected expression was not from tracheae but truly from non-epidermal cells, they should explain about that. If not, they should state that the possibility of detection of CP gene expression from trachea in the non-epidermal tissues was not ruled out. Expression in hemolymph might be OK, because cells in hemolymph are isolated. The expression of CP genes in hemolymph is interesting.

Response 2: Addressed. Thanks for your critical suggestion. We did not have a special technique to remove tracheae completely from tissues including prothoracic glands and corpora allata. Please see Line 428-442.

Our study provided the transcriptome-based expression profiles of CP genes in the larval corpus allatum, prothoracic gland and haemolymph. The expression of CP genes in the corpus allatum and prothoracic gland was observed in the larval stage of B. mori where corpus allatum and prothoracic gland all are separated organs. Generally, previous studies focused on CPs by investigating the transcription profiles in the larval epidermis as well as the non-epidermis tissues/organs [7, 25, 28, 44]. So far, in insects, CP genes have been found to be expressed in a large number of non-epidermis tissues/organs, such as nervous system, antenna, fat body, ovary or testis, brain, silk gland, male accessory gland, midgut, muscle [7, 10, 25, 28]. However, one should be aware that contamination of tracheal cells during the dissection of the tissues might result in false positive results [7]. Trachea can penetrate most of the tissues including prothoracic gland and corpora allatum, which also might result in false positive results. Thus, we have to state that the possibility of detection of CP gene expression from trachea in the non-epidermal tissues was not ruled out. Our work provided evidence for the expression of CP genes in the haemolymph, where the expression of CP genes is interesting. The CPs synthesized in the non-epidermis cells perhaps make use of them in these organs [45]. This is supported by a study in B. mori, where a CPT1 gene is detected in the fat body and has proved to participate in the silkworm innate immunity by recognition of Escherichia coli [41]. Since CP distribution are highly diverse among not only body regions but also developmental stages, the whole expression profile of CP genes in the epidermis of a certain insect species needs to be further addressed in the future.

Point 3: Line 451, trachea as a non-epidermis tissue. “The tracheae are invaginations of the epidermis and thus their lining is continuous with the body cuticle” (cited from P. J. Gullan and P. S. Cranston, The Insect: an outline of entomology, Wiley Blackwell). There is also similar explanation in another textbook (Marc J. Klowden, Physiological Systems in Insects, Academic Press). It is not appropriate to treat trachea as a non-epidermis tissue.

Response 3: Addressed.

Thank you for your comment. The related “trachea” and sentences have been deleted. Please see Line 432 and 437-439.

Point 4: Line 18, two silkworms. “Two silkworms” sounds like two individuals of silkworm. This should be like “two silkworm species”. The same expression is seen at line 250.

Response 4: Addressed.